# ORTHONORMAL REGULARIZATION IN LOW-RANK ADAPTATION

## ABSTRACT

Performance degradation on tasks outside the fine-tuning domain is often observed while performing parameter-efficient fine-tuning (PEFT) on neural networks with limited data. For example, fine-tuning on mathematical datasets may impair the large language model's coding ability. We analyze this issue and identify the condition number of weight matrices as a key factor contributing to such degradation. To address this, we propose Singular Values and Orthonormal Regularized Singular Vectors Adaptation, or SORSA, a novel PEFT method that explicitly improves the conditioning of the adapted model parameters, thereby mitigating degradation and preserving broader capabilities. Empirically, we demonstrate that SORSA outperforms full fine-tuning, LoRA, PiSSA and AdaLoRA.

## 1 INTRODUCTION

Pre-trained large language models (LLMs) demonstrate strong generalization capabilities, enabling them to perform a wide range of natural language processing (NLP) tasks (Brown et al., 2020; Achiam et al., 2023; Touvron et al., 2023; Peng et al., 2024; Grattafiori et al., 2024). For adapting LLMs to specific downstream tasks, the default approach is often full parameter fine-tuning (Full FT), which updates all model parameters.

However, as LLMs continue to grow in scale, Full FT becomes increasingly impractical due to high computational and memory demands. To alleviate this, Parameter-Efficient Fine-Tuning (PEFT) methods have gained popularity, offering a cost-effective alternative by only updating a small subset of parameters.

Among PEFT approaches, LoRA (Hu et al., 2022) has emerged as a preferred choice due to its simplicity, efficiency, and minimal impact on inference-time latency. LoRA injects low-rank trainable matrices into the model, enabling effective fine-tuning with significantly reduced resource requirements.

Despite its efficiency, LoRA and similar PEFT methods face a major challenge under low-data regimes: they tend to overfit and degrade the model's original generalization ability, and even cause catastrophic forgetting (Xu et al., 2021a; Lin et al., 2024; Shuttleworth et al., 2024; van de Ven et al., 2024). For instance, fine-tuning on a small mathematical dataset may cause the model to forget previously acquired capabilities such as code generation or commonsense reasoning.

Previous works (Sinha et al., 2018; Saratchandran et al., 2024; Feng et al., 2025) have shown that neural networks with well-conditioned weight is able to provide a more robust performance. We further analyze this phenomenon in the context of PEFT, and identify the condition number of weight matrices as a critical factor affecting generalization during fine-tuning. Our study shows that LoRA often amplifies the condition number, making the adapted model increasingly ill-conditioned and unstable.

To address this, we propose a new PEFT method that explicitly improves the conditioning of the model during training. Our approach introduces orthonormal regularization to maintain well-conditioned weights, thereby preserving the model's generalization while enabling efficient adaptation. Empirical results show that our method significantly mitigates overfitting and outperforms existing baselines across various tasks.

We summarize our main contributions as follows:

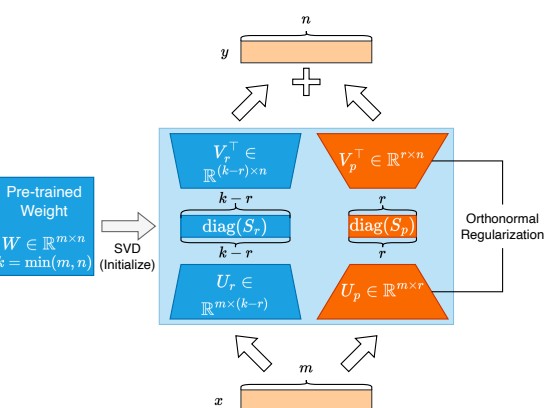

Figure 1: Illustration of SORSA.

- We demonstrate that during PEFT, well-conditioned weights tend to have better generalization.
- We propose SORSA, a novel parameter-efficient fine-tuning (PEFT) method that combines low-rank SVD-based initialization with orthonormal regularization.
- We provide the convergence rate of SORSA with gradient descent. (Theorem 5.4)
- We provide theoretical analysis showing that the orthonormal regularizer leads to better-conditioned weight updates. (Theorem 5.6)
- We empirically demonstrate that SORSA consistently outperforms or matches the performance of strong baselines, including full fine-tuning, LoRA, PiSSA, and AdaLoRA.

**Roadmap.** In Section 2, we present related work. In Section 3, we introduce the preliminary for our work. In Section 4, we propose our PEFT method. In Section 5, we provide theoretical analysis for SORSA. In Section 6, we conduct extensive experiments to validate SORSA's capability. In Section 7, we provide discussion and conclude the paper.

## 2 RELATED WORK

**Efficient Computation in Machine Learning.** As the increasing scale of training data and model parameters, developing efficient machine learning algorithms have become central focus of recent AI research. In visual recognition, the acceleration of CNN (O'shea & Nash, 2015; He et al., 2016) and ViT (Dosovitskiy et al., 2020) have long been a heated topic, especially for edge devices that have limited computation resources. Representative acceleration techniques including architectural simplification (Sandler et al., 2018; Ding et al., 2021), quantization (Wu et al., 2016; Liu et al., 2021), and pruning (Yu et al., 2022). These techniques have significantly advance in real world applications, e.g. autonomous driving (Jiang et al., 2023b), medical image segmentation (Han et al., 2022), remote sensing (Xu et al., 2021b), emotion recognition (Zhang et al., 2021; Zhao et al., 2021; Liu et al., 2022a), and industrial automation. In content creation, diffusion models (Ho et al., 2020; Rombach et al., 2022) and flow matching models (Lipman et al., 2022; Liu et al., 2022c) are high-fidelity visual content generators. Acceleration in this area focuses on model architecture design (Dao et al., 2023; Frans et al., 2024; Chen et al., 2025; Cao et al., 2025a), fast ODE sampler (Xue et al., 2024b), complexity analysis (Gupta et al., 2024; Ke et al., 2025), distillation (Meng et al., 2023). These works have inspired many future applications, e.g. education, drug discovery (Wen et al., 2024), face synthesis (Liu et al., 2022b), and advertising (Liu et al., 2024a), and directions, e.g. benchmarks (Cao et al., 2025b; Guo et al., 2025a;b;c) and theoretical explorations (Hu et al., 2024). Graph Neural Networks are fundamental tools to model complex relational data (Veličković et al., 2018; Xu et al., 2019; Li et al., 2025), where important acceleration techniques include sparsification (Morris et al., 2020; Liu et al., 2023), GNN to MLP distillation (Zhang et al., 2022a; Han et al., 2023), and lazy computation (Narayanan et al., 2022; Zhang et al., 2024; Xue et al., 2024a). These techniques has inspired applications including but not limited to spatio-temporal data mining (Zhang et al., 2022b;

Wang et al., 2022), fake news detection (Xu et al., 2022; Chang et al., 2024), human skeleton-based visual recognition (Li et al., 2021; Fu et al., 2021), while also inspired aspects of graph neural networks including mitigating sensitive data influence (Chien et al., 2023; Zhang, 2024; Yi & Wei, 2025), and robustness (Geisler et al., 2021; Deng et al., 2022).

**PEFT Methods.**    PEFT methods have been proposed to alleviate the inefficiency of full-parameter fine-tuning for large language models. These methods update only a small subset of parameters, often keeping the majority of the pre-trained model frozen, which significantly reduces memory and computational costs during training.

Adapter-based approaches were among the earliest PEFT methods, introduced by (Houlsby et al., 2019), where small trainable modules are inserted between frozen layers. Subsequent works such as (Lin et al., 2020) and (He et al., 2021) explored more compact or parallelized adapter designs. However, all adapter-based methods generally incur additional inference-time latency, since the inserted modules are not mergeable with the original model weights.

LoRA (Hu et al., 2022) gained popularity for introducing low-rank trainable matrices added to the pre-trained weight matrices. This approach avoids inference latency while offering competitive performance. Variants of LoRA expand upon this idea: AdaLoRA (Zhang et al., 2023) improves parameter efficiency by incorporating dynamic rank selection via singular value decomposition and pruning. DoRA (Liu et al., 2024b) decouples the direction and magnitude of weight updates, achieving higher expressiveness at the cost of higher training-time computation. OLoRA (Büyükakyüz, 2024) uses orthogonal initialization via QR decomposition to improve convergence speed. PiSSA (Meng et al., 2024) decomposes the pre-trained weight matrix and isolates a residual component, which remains frozen during training to improve convergence and stability.

Prompt-based PEFT methods, such as prefix-tuning (Lester et al., 2021), prepend learnable tokens to the model input. Although these methods are simple to implement, they often lead to longer input sequences and require careful prompt engineering. Other recent advances include GaLore (Zhao et al., 2024), which reduces memory usage through low-rank gradient accumulation, and LISA (Pan et al., 2024), which selectively fine-tunes critical layers using layer-wise importance sampling.

**Condition Numbers in Neural Networks**

## 3    PRELIMINARY

In this section, we first introduce our notations, then provide preliminary for our work.

### 3.1    NOTATIONS

We used $\mathbb{R}$ to denote set of real numbers. We use $A \in \mathbb{R}^{n \times d}$ to denote an $n \times d$ size matrix where each entry is a real number. We use $I_d$ to denote the $d \times d$ identity matrix. We use $A^\top$ to denote the transpose of a matrix $A$. We use $A^{1/2}$ to denote element-wise square root of the matrix $A$, i.e. $(A^{1/2})_{i,j} = (A_{i,j})^{1/2}$. We use $\|A\|_F$ to denote Frobenius norm of matrix $A$. We use $\|A\|$ to denote spectral norm of matrix $A$. We use $A \preceq B$ to denote the positive semidefinite order, i.e. for symmetric $A, B \in \mathbb{R}^{d \times d}$, $A \preceq B \iff B - A \succeq 0$.

### 3.2    PEFT METHODS

**LoRA**    LoRA (Hu et al., 2022) represents the weight as a low-rank decomposition:

$$W = W_0 + BA,$$

where $W_0 \in \mathbb{R}^{m \times n}$ is the frozen pre-trained weight, $A \in \mathbb{R}^{m \times r}$ is Gaussian-initialized, and $B \in \mathbb{R}^{r \times n}$ is initialized with zeros.

**AdaLoRA.**    AdaLoRA (Zhang et al., 2023) introduces dynamic rank adaptation via SVD, and prunes less significant singular values to reduce parameter overhead.

**DoRA.** DoRA (Liu et al., 2024b) reformulates the weight update as a normalized decomposition:

$$W = m \cdot \frac{W_0 + BA}{\|W_0 + BA\|_c},$$

where $m = \|W_0 + BA\|_c$ is the column-wise norm. This improves model capacity but increases computational cost per step.

**OLoRA.** OLoRA (Büyükakyüz, 2024) initializes $A$ and $B$ using QR decomposition, ensuring orthonormality in the initial adapter weights, which empirically speeds up convergence.

**PiSSA.** PiSSA (Meng et al., 2024) decomposes $W_0$ via SVD as $W_0 = U\Sigma V^\top$ and splits it into:

$$W_{\text{pri}} = AB, \quad \text{where } A = U_p S_p^{1/2}, \quad B = S_p^{1/2} V_p^\top,$$

with $U_p, S_p, V_p$ being the top-$r$ components. The residual $W_{\text{res}} = U_r S_r V_r^\top$ remains frozen during training. This results in faster convergence and improved model fit.

### 3.3 CONDITION NUMBER

We here provide a formal definition for the condition number.

**Definition 3.1** (Condition Number). *Let $A \in \mathbb{R}^{m \times n}$ be a matrix with full column rank. The* condition number *of $A$ with respect to the spectral norm is defined as*

$$\kappa(A) := \frac{\sigma_{\max}(A)}{\sigma_{\min}(A)} = \|A\| \cdot \|A^{-1}\|,$$

*where $\sigma_{\max}(A)$ and $\sigma_{\min}(A)$ are the largest and smallest nonzero singular values of $A$.*

## 4 OUR METHOD

Giving a matrix $W \in \mathbb{R}^{m \times n}$, with $m \geq n$ (without loss of generality), we could perform SVD to decompose $W$ by $W = U \operatorname{diag}(S) V^\top$. Here, $U \in \mathbb{R}^{m \times k}$ is a matrix of left singular vectors and has orthonormal columns, $V \in \mathbb{R}^{n \times k}$ is a matrix of right singular vectors and has orthonormal columns, and $S \in \mathbb{R}^k$ are singular values $\sigma^1, \sigma^2 \ldots \sigma^k$ arranged in descending order. $\operatorname{diag}(S)$ is constructed by placing the elements of $S \in \mathbb{R}^k$ along the main diagonal, with all other elements zero.

According to our SVD notations, given a rank $r$ where $r \ll k$, we could perform the low-rank approximation by selecting the first $r$ items on the diagonal of $\Sigma$, which is the first $r$ most significant singular values, and also select the first $r$ columns of $U$ and first $r$ rows of $V^\top$, which correspond to the selected singular values. By performing SVD low-rank approximation, we could get a low-rank matrix that preserves the largest significant values and vectors, containing the matrix's "most essential" data.

We use $\Sigma_p \in \mathbb{R}^{n \times n}$ to denote a diagonal matrix where first $r$ entries are non-zero and all the remaining $n - r$ entries. Similarly, we use $\Sigma_r \in \mathbb{R}^{n \times n}$ to denote a diagonal matrix where first $n - r$ entries are non-zero and all the remaining $r$ entries are zeros. Let $\Sigma = \Sigma_p + \Sigma_r$. Let SVD of $W$ be $W = U\Sigma V^\top$.

Therefore, for a pre-trained weight $W_0 \in \mathbb{R}^{m \times n}$, we could split it based on its singular value into principal weight $W_p$ and residual weight $W_r$,

$$W_p := \underbrace{U}_{m \times n} \underbrace{\Sigma_p}_{n \times n} \underbrace{V^\top}_{n \times n} \in \mathbb{R}^{m \times n}, \quad W_r := \underbrace{U}_{m \times n} \underbrace{\Sigma_r}_{n \times n} \underbrace{V^\top}_{n \times n} \in \mathbb{R}^{m \times n}.$$

Here, $U$ represents the matrix of left singular vectors, $S$ represents the singular values, $\operatorname{diag}(W)$ denotes a function to form a diagonal matrix from $W$, and $V$ represents the matrix of right singular vectors. Since $\Sigma_p$ is zeroed out in the last $n - r$ entries, and $\Sigma_r$ is zeroed out in the first $r$ entries, we can easily find low-rank equivalents of $W_p$ and $W_r$. Specifically,

$$W_p = \underbrace{U_p}_{m \times r} \underbrace{S_p}_{r \times r} \underbrace{V_p^\top}_{r \times n}, \quad W_r = \underbrace{U_r}_{m \times (n-r)} \underbrace{S_r}_{(n-r) \times (n-r)} \underbrace{V_r^\top}_{(n-r) \times n},$$

where $U_p$ is the first $r$ columns of $U$, $S_p$ is the first $r$ columns and rows of $\Sigma_p$, $V_p$ is the first $r$ columns of $V$, $U_r$ is the last $n - r$ columns of $U$, $S_r$ is the last $n - r$ columns and rows of $\Sigma_r$, $V_r$ is the last $n - r$ column of $V$.

The initialization of $W_r$ in SORSA is same as PiSSA (Meng et al., 2024). Nevertheless, unlike PiSSA which merge $S_p$ with $U_p$ and $V_p^\top$ into $A$ and $B$ by $A = U_p S_p^{1/2}$ and $B = S_p^{1/2} V_p^\top$, SORSA remains $U_p$, $S_p$, and $V_p^\top$ in separate weight. SORSA is defined by Eq. (1), initially equivalent to the pre-trained weight $W_0$. During training, $W_r$ remains frozen, and only $U_p$, $S_p$, and $V_p^\top$ are updated.

SORSA is defined as:

$$\text{SORSA}(x) := x(W_r + W_p) = xW_r + xU_p \operatorname{diag}(S_p)V_p^\top. \tag{1}$$

We adopt an orthonormal regularizer for $U_p$ and $V_p$.

**Definition 4.1** (Orthonormal regularizer). *The orthonormal regularizer is defined as*

$$\mathcal{L}_{\text{reg}}(U_p, V_p) := \|U_p^\top U_p - I_m\|_F^2 + \|V_p^\top V_p - I_n\|_F^2.$$

The regularizer could enhance their orthonormality during training. We discuss and verify its importance and effectiveness in 5.

Therefore, parameter updating of $W_p$ in a SORSA adapter at training step $t$ could be expressed as:

$$W_{p,t+1} = W_{p,t} - \eta_t \nabla_{W_{p,t}} \mathcal{L}_{\text{train}} - \gamma_t \nabla_{W_{p,t}} \mathcal{L}_{\text{reg}}. \tag{2}$$

At training step $t$, $\nabla_{W_{p,t}} \mathcal{L}_{\text{train}}$ denotes the gradient of $\mathcal{L}_{\text{train}}$ respect to $W_{p,t}$, and $\nabla_{W_{p,t}} \mathcal{L}_{\text{reg}}$ denotes the gradient of the orthonormal regularizer loss $\mathcal{L}_{\text{reg}}$ respect to $W_{p,t}$. $\eta_t$ and $\gamma_t$ are the learning rates for training loss and regularizer loss at step $t$, respectively.

We update the SORSA as the following for implementation simplicity

$$W_{p,t+1} = W_{p,t} - \eta_t \left( \nabla_{W_{p,t}} \mathcal{L}_{\text{train}} + \frac{\gamma}{\eta_d} \nabla_{W_{p,t}} \mathcal{L}_{\text{reg}} \right), \tag{3}$$

$\eta_d$ is the maximum learning rate from the scheduler. This implementation allows us to use only one optimizer and scheduler to deal with two different learning rates separately.

# 5 THEORETICAL ANALYSIS

## 5.1 CONVERGENCE RATE

We begin by analyzing the convergence behavior of gradient descent when applied to our objective function, which consists of a data-fitting loss $L_{\text{train}}$ and our orthonormal regularizer $\mathcal{L}_{\text{reg}}$.

**Lemma 5.1** (Lipschitz continuity of $\mathcal{L}_{\text{reg}}$). *Suppose $\|U_p\|_F \leq M_U$ and $\|V_p\|_F \leq M_V$. Then $\mathcal{L}_{\text{reg}}$ is Lipschitz continuous in the Frobenius norm:*

$$|\mathcal{L}_{\text{reg}}(U_p^1, V_p^1) - \mathcal{L}_{\text{reg}}(U_p^2, V_p^2)| \leq L_{\text{reg}}(\|U_p^1 - U_p^2\|_F + \|V_p^1 - V_p^2\|_F),$$

*where*

$$L_{\text{reg}} = 4M_U(M_U^2 + 1) + 4M_V(M_V^2 + 1).$$

*Proof.* Compute the partial gradients

$$\nabla_{U_p} \mathcal{L}_{\text{reg}} = 4U_p(U_p^\top U_p - I_m),$$
$$\nabla_{V_p} \mathcal{L}_{\text{reg}} = 4V_p(V_p^\top V_p - I_n).$$

Hence

$$\|\nabla \mathcal{L}_{\text{reg}}\|_F \leq 4\|U_p\|\|U_p^\top U_p - I_m\|_F + 4\|V_p\|\|V_p^\top V_p - I_n\|_F$$
$$\leq 4M_U(M_U^2 + 1) + 4M_V(M_V^2 + 1).$$

By the mean value theorem for vector functions,

$$|\mathcal{L}_{\text{reg}}(X) - \mathcal{L}_{\text{reg}}(Y)| \leq \max_Z \|\nabla \mathcal{L}_{\text{reg}}(Z)\|_F \|X - Y\|_F,$$

and the claimed bound follows. $\square$

We now make two standard assumptions to ensure well-behaved optimization.

**Assumption 5.2** (Smoothness and strong convexity of $L_{\text{train}}$). *The data term $L_{\text{train}}(W_p)$ is twice differentiable, $\mu_{\text{train}}$-strongly convex and $L_{\text{train}}$-smooth:*

$$\mu_{\text{train}}I \preceq \nabla^2 L_{\text{train}}(W) \preceq L_{\text{train}}I \quad \text{for all } W_p.$$

**Assumption 5.3** (Hessian lower bound for $\mathcal{L}_{\text{reg}}$). *There is a constant $C_{\text{reg}} \geq 0$ such that*

$$\nabla^2 \mathcal{L}_{\text{reg}}(W) \succeq -C_{\text{reg}}I \quad \text{for all } W = (U_p, V_p).$$

The next theorem establishes that, under these assumptions, SORSA converges linearly.

**Theorem 5.4** (Linear convergence of SORSA). *Let*

$$F(W_p) = L_{\text{train}}(W_p) + \gamma \mathcal{L}_{\text{reg}}(W_p),$$

*and suppose Assumptions 5.2 and 5.3 hold. If*

$$0 < \gamma < \frac{\mu_{\text{train}}}{C_{\text{reg}}}, \quad \eta \in (0, \frac{2}{L_{\text{train}} + \gamma L_{\text{reg}}}),$$

*then gradient descent*

$$W_p^{t+1} = W_p^t - \eta \nabla F(W_p^t)$$

*satisfies*

$$F(W_p^t) - F(W_p^*) \leq (1 - \eta(\mu_{\text{train}} - \gamma C_{\text{reg}}))^t (F(W_p^0) - F(W_p^*)).$$

*In particular, setting $\eta = 1/(L_{\text{train}} + \gamma L_{\text{reg}})$ gives*

$$F(W_p^t) - F(W_p^*) \leq (1 - \frac{\mu_{\text{train}} - \gamma C_{\text{reg}}}{L_{\text{train}} + \gamma L_{\text{reg}}})^t (F(W_p^0) - F(W_p^*)).$$

*Proof.* By Assumption 5.2, $\nabla^2 L_{\text{train}} \geq \mu_{\text{train}}I$ and by Assumption 5.3, $\nabla^2(\gamma \mathcal{L}_{\text{reg}}) \geq -\gamma C_{\text{reg}}I$. Hence,

$$\nabla^2 F = \nabla^2 L_{\text{train}} + \gamma \nabla^2 \mathcal{L}_{\text{reg}} \geq (\mu_{\text{train}} - \gamma C_{\text{reg}})I,$$

and also $\nabla^2 F \leq (L_{\text{train}} + \gamma L_{\text{reg}})I$. The claimed rate follows from standard gradient descent guarantees. □

## 5.2 CONDITION NUMBER

We now analyze how the regularizer in SORSA helps maintain a smaller condition number for the weight matrix. A well-conditioned weight matrix is essential for stable optimization and good generalization.

We begin with a lemma that shows the singular values of the regularized weight matrix stay close to those of the unregularized one, provided the regularizer gradient is small.

**Lemma 5.5.** *Let*

$$W_p^{\text{unreg},t} = U_p^{\text{unreg},t} S_p^{\text{unreg},t} (V_p^{\text{unreg},t})^\top, \quad W_p^{\text{reg},t} = U_p^{\text{reg},t} S_p^{\text{reg},t} (V_p^{\text{reg},t})^\top$$

*be the outputs of one step of SORSA at step $t$ with and without regularizer, respectively.*

*If $\|\nabla_{W_p} \mathcal{L}_{\text{reg}}\|_F \leq \epsilon_\nabla$, then for each singular value $\sigma_i$,*

$$(1 - \epsilon)\sigma_i^{\text{unreg},t} \leq \sigma_i^{\text{reg},t} \leq (1 + \epsilon)\sigma_i^{\text{unreg},t},$$

*where $\epsilon = \gamma \epsilon_\nabla$.*

*Proof.* We have

$$W_p^{\text{reg}} - W_p^{\text{unreg},t} = \gamma \nabla_{W_p} \mathcal{L}_{\text{reg}}, \quad \|W_p^{\text{reg}} - W_p^{\text{unreg},t}\|_F = \gamma \epsilon_\nabla.$$

By Weyl's inequality,

$$|\sigma_i^{\text{reg},t} - \sigma_i^{\text{unreg},t}| \leq \|W_p^{\text{reg},t} - W_p^{\text{unreg},t}\| \leq \|W_p^{\text{reg},t} - W_p^{\text{unreg},t}\|_F \leq \gamma \epsilon_\nabla.$$

The last inequality follows directly. □

We now prove our main theorem: the condition number of the regularized weight matrix is strictly smaller than that of the unregularized one.

**Theorem 5.6.** *Under the setup of Lemma 5.5, assume that $\nabla\mathcal{L}_{\text{train}}$ is invariant for all $t > 0$. Let the orthonormal regularizer be defined in Definition 4.1 Then for every iteration $t > 0$,*

$$\kappa(W_p^{\text{reg},t}) < \kappa(W_p^{\text{unreg},t}),$$

*where $\kappa$ is defined in Definition 3.1.*

*Proof.* We divide the proof into four steps to illustrate how regularization improves conditioning.

**Step 1. Factor-wise bounds.** For any factorization $W = USV^\top$ with diagonal $S$,

$$\|W\| \le \|U\|\|S\|\|V\|, \quad \|W^{-1}\| \le \|V\|\|S^{-1}\|\|U^{-1}\|.$$

Hence,

$$\kappa(W) \le \kappa(U)\kappa(S)\kappa(V).$$

**Step 2. Singular value perturbation.** According to Lemma 5.5,

$$|\sigma_i^{\text{reg},t} - \sigma_i^{\text{unreg},t}| \le \epsilon_t,$$

which implies

$$\kappa(S_p^{\text{reg},t}) \le \frac{1+\epsilon_t}{1-\epsilon_t}\kappa(S_p^{\text{unreg},t}).$$

**Step 3. Orthonormal regularizer bounds factor condition numbers.** By definition of $\mathcal{L}_{\text{reg}}$ in Definition 4.1, and $\nabla\mathcal{L}_{\text{train}}$ is invariant for all $t > 0$,

$$\kappa(U_p^{\text{reg},t}) < \kappa(U_p^{\text{unreg},t}), \quad \kappa(V_p^{\text{reg},t}) < \kappa(V_p^{\text{unreg},t}).$$

**Step 4: Combine bounds to compare condition numbers.** By the above,

$$\kappa(W_p^{\text{reg},t}) \le \kappa(U_p^{\text{reg},t})\kappa(S_p^{\text{reg},t})\kappa(V_p^{\text{reg},t})$$
$$\le \kappa(U_p^{\text{reg},t})\kappa(V_p^{\text{reg},t})\frac{1+\epsilon_t}{1-\epsilon_t}\kappa(S_p^{\text{unreg},t}),$$

and

$$\kappa(W_p^{\text{unreg},t}) \ge \kappa(U_p^{\text{unreg},t})\kappa(S_p^{\text{unreg},t})\kappa(V_p^{\text{unreg},t}).$$

So,

$$\frac{\kappa(W_p^{\text{reg},t})}{\kappa(W_p^{\text{unreg},t})} \le \frac{\kappa(U_p^{\text{reg},t})\kappa(V_p^{\text{reg},t})}{\kappa(U_p^{\text{unreg},t})\kappa(V_p^{\text{unreg},t})} \cdot \frac{1+\epsilon_t}{1-\epsilon_t} < 1.$$

Thus, $\kappa(W_p^{\text{reg},t}) < \kappa(W_p^{\text{unreg},t})$, completing the proof. □

# 6 EXPERIMENTS

We conducted comparative experiments on different NLP tasks, including natural language generation (NLG) between SORSA, PiSSA (Meng et al., 2024), LoRA (Hu et al., 2022), AdaLoRA (Zhang et al., 2023), and full parameter fine-tuning.

We conducted NLG tests on Llama 2 7B (Touvron et al., 2023), RWKV6 7B (Peng et al., 2024), Mistral 7B v0.1 (Jiang et al., 2023a) and Gemma 7B (Gemma Team, 2024). We trained the models using the first 100K data in MetaMathQA (Yu et al., 2023) and evaluated the model on GSM-8K (Cobbe et al., 2021) and MATH (Hendrycks et al., 2021). We also trained the model on the first 100K data in CodeFeedback Filtered Instruction (Zheng et al., 2024) dataset and evaluated it on HumanEval (Chen et al., 2021). The training process followed identical setups as the experiments conducted in PiSSA (Meng et al., 2024). All reported values are accuracy in percentage. See

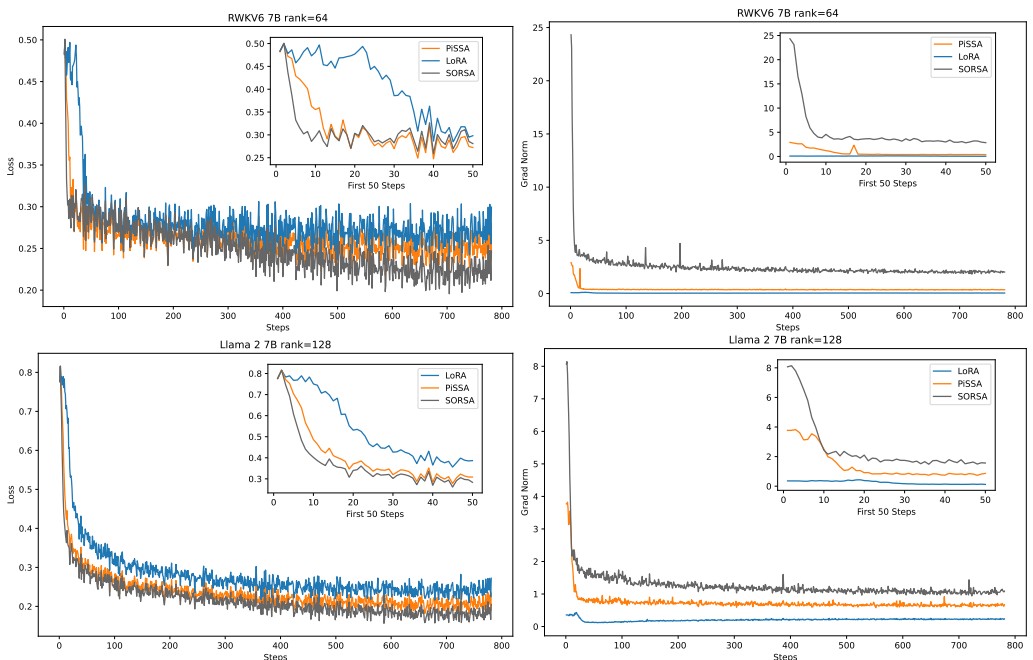

Figure 2: The training loss and gradient norm comparison between SORSA, PiSSA, and LoRA on MetaMathQA training of RWKV6 7B and Llama 2 7B. LoRA and PiSSA curves of Llama 2 7B are from (Meng et al., 2024).

Table 1: Comparing SORSA with other methods on NLG tasks. † denotes results from (Meng et al., 2024). We use **TPara.** to represent trainable parameters.

| Model | Method | TPara. | GSM-8K | MATH | HumanEval |
|-------|--------|--------|--------|------|-----------|
| Llama 2 7B | Full FT | 6738M | 49.05† | 7.22† | 21.34† |
| Llama 2 7B | LoRA | 320M | 42.30† | 5.50† | 18.29† |
| Llama 2 7B | PiSSA | 320M | 53.07† | 7.44† | 21.95† |
| Llama 2 7B | AdaLoRA | 320M | 47.30 | 6.48 | 19.51 |
| Llama 2 7B | SORSA | 320M | **56.03** | **10.36** | **24.39** |
| RWKV6 7B | LoRA | 176M | 8.04] | 7.38 | 15.24 |
| RWKV6 7B | PiSSA | 176M | 32.07 | 9.42 | 17.07 |
| RWKV6 7B | AdaLoRA | 176M | 33.28 | 8.08 | 15.85 |
| RWKV6 7B | SORSA | 176M | **45.87** | **11.32** | **22.56** |
| Mistral 7B | Full FT | 7242M | 67.02† | 18.60† | 45.12† |
| Mistral 7B | LoRA | 168M | 67.70† | 19.68† | 43.90† |
| Mistral 7B | PiSSA | 168M | 72.86† | 21.54† | 46.95† |
| Mistral 7B | AdaLoRA | 168M | 72.25 | 21.06 | 45.73 |
| Mistral 7B | SORSA | 168M | **73.09** | **21.86** | **47.56** |
| Gemma 7B | Full FT | 8538M | 71.34† | 22.74† | 46.95† |
| Gemma 7B | LoRA | 200M | 74.90† | 31.28† | 53.66† |
| Gemma 7B | PiSSA | 200M | 77.94† | **31.94** † | 54.27† |
| Gemma 7B | AdaLoRA | 200M | **78.99** | 31.44 | **55.49** |
| Gemma 7B | SORSA | 200M | 78.09 | 29.52 | **55.49** |

Section A for more details and hyperparameters of the training. We quoted some PiSSA, LoRA, and full parameter fine-tuning results from (Meng et al., 2024). Some of our experiments were conducted on a single NVIDIA A100-SXM4 (80GB) GPU, and others were conducted on a single NVIDIA H100-SXM4 (80GB) GPU. See Table 1 for the results and Figure 2 for the loss and gradient norm comparison.

The results showed that across all models tested, SORSA generally outperformed other methods, though with some notable exceptions. For mathematical evaluations on Llama 2 7B, SORSA scored 56.03% on GSM-8K and 10.36% on MATH, significantly outperforming other methods. For the RWKV6 7B model, SORSA achieved 45.87% accuracy on GSM-8K and 11.32% on MATH, surpassing both PiSSA and AdaLoRA, with AdaLoRA showing competitive performance on GSM-8K at 33.28%. On Mistral 7B, SORSA reached 73.09% on GSM-8K and 21.86% on MATH, showing modest improvements over AdaLoRA's strong performance of 72.25% and 21.06%, respectively. With Gemma 7B, the results were mixed - while AdaLoRA achieved the highest GSM-8K score at 78.99% and competitive MATH performance at 31.44%, SORSA maintained strong performance with 78.09% on GSM-8K. However, its MATH score of 29.52% was lower than other methods. In coding evaluations, SORSA and AdaLoRA showed strong performance on HumanEval, with both methods achieving 55.49% on Gemma 7B, while SORSA maintained an edge across other model variants. Additionally, we did not include loss and gradient norm curves in our figure because the regularizer in AdaLoRA and Gaussian initialization caused significantly higher initial loss values, making direct comparisons with other methods inappropriate.

The Figure 2 reveals that SORSA and PiSSA exhibit nearly identical loss curves at the beginning and even slightly higher than PiSSA on RWKV-6 training. However, when the training step is approximately $t > 300$, SORSA steadily decreases its loss. In contrast, LoRA and PiSSA show a deceleration in their loss reduction. The observations on loss curves are also valid for the changing rate of gradient norm, where SORSA showed a more consistent decrease in gradient norm compared to LoRA and PiSSA. This can be explained by Theorem 5.6, especially at later stages of training.

## 7  DISCUSSION AND CONCLUSION

In this paper, we introduced SORSA, a novel parameter-efficient fine-tuning (PEFT) method designed to enhance the adaptation of large language models (LLMs) for downstream tasks. SORSA utilizes singular value decomposition (SVD) to split pre-trained weights into principal and residual components, only training the principal singular values and vectors while freezing the residuals. We implemented an orthonormal regularizer to maintain the orthonormality of singular vectors during training, ensuring efficient parameter updates and preserving the integrity of singular values.

Our experiments demonstrated that SORSA outperforms existing PEFT methods, such as LoRA and PiSSA, in both convergence speed and accuracy on the NLG tasks. Specifically, Llama 2 7B, tuned with SORSA, achieved significant improvements in the GSM-8K and MATH benchmarks, highlighting the effectiveness of our approach.

We adopted singular values and vector analysis, comparing SORSA with FT and LoRA. SORSA is superior in preserving the pre-trained weight's singular values and vectors during training. This suggests an explanation for SORSA's supreme performance demonstrated in the experiment. We also show the significance of the orthonormal regularizer through analysis.

Our theoretical analysis provided a mathematical foundation for SORSA, demonstrating its convexity, Lipschitz continuity, and the crucial role of the regularizer in improving the optimization landscape. This theoretical framework explains SORSA's empirical superior performance and offers valuable insights for future developments in adaptive learning algorithms.

SORSA retains the advantages of LoRA and variants, including low training VRAM requirements, no inference latency, and versatility across different neural network architectures. By offering a more efficient fine-tuning mechanism, SORSA presents a promising direction for future research and application in the field of LLMs.

Overall, SORSA gives a new perspective on parameter-efficient fine-tuning, showcasing exceptional efficiency and robust performance. It outperforms existing methods like LoRA and PiSSA in several downstream tasks and maintains the practical benefits of low VRAM requirements, no inference latency, and ease of implementation. This innovative approach offers a promising direction of singular values and vector analysis for future research and practical applications in adapting pre-trained models, making it a pivotal development in the field.

---

[1]This significant under-perform due to LoRA failed to learn the GSM-8K required answer formatting behavior.

## ETHIC STATEMENT

This paper does not involve human subjects, personally identifiable data, or sensitive applications. We do not foresee direct ethical risks. We follow the ICLR Code of Ethics and affirm that all aspects of this research comply with the principles of fairness, transparency, and integrity.

## REPRODUCIBILITY STATEMENT

We ensure reproducibility on both theoretical and empirical fronts. For theory, we include all formal assumptions, definitions, and complete proofs in the appendix. For experiments, we describe model architectures, datasets, preprocessing steps, hyperparameters, and training details in the main text and appendix. Code and scripts are provided in the supplementary materials to replicate the empirical results.

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

# Appendix

**Roadmap.** In the appendix, we present the experiments details in Section A.

## A    EXPERIMENTS DETAILS

For our NLG tasks, we adapted Llama 2 7B (Touvron et al., 2023), RWKV6 7B (Peng et al., 2024), Mistral 7B v0.1 (Jiang et al., 2023a) Gemma 7B (Gemma Team, 2024) models by SORSA. For GSM-8K (Cobbe et al., 2021) and MATH (Hendrycks et al., 2021) evaluations, we trained those models with the first 100K data in MetaMathQA (Yu et al., 2023) dataset. For HumanEval (Chen et al., 2021) evaluation, we use the first 100K data in CodeFeedback Filtered Instruction (Zheng et al., 2024) dataset.

We used AdamW (Loshchilov & Hutter, 2017) optimizer and cosine annealing scheduler in training. SORSA adapters were applied on all linear matrices in every layer. We only calculated the loss on the response part. The models are loaded in FP32 and trained with TF32 & BF16 mix precision. In our experiments, we selected a higher learning rate for SORSA than other methods to counterbalance the negative effect of orthonormal regularizer on optimizing toward lower training loss. See Table 2 and 3 for hyperparameters.

Table 2: Hyperparameters for training with SORSA, LoRA and PiSSA on different models for GSM-8K and MATH

| Model | Llama 2 7B | RWKV6 7B | RWKV6 7B | Mistral 7B | Gemma 7B |
|---|---|---|---|---|---|
| Method | SORSA | SORSA | LoRA&PiSSA | SORSA | SORSA |
| Mix-Precision | TF32+BF16 | TF32+BF16 | TF32+BF16 | TF32+BF16 | TF32+BF16 |
| Epoch | 1 | 1 | 1 | 1 | 1 |
| Batch Size | 128 | 128 | 128 | 128 | 128 |
| Max Length | 512 | 512 | 512 | 512 | 512 |
| Weight Decay | 0 | 0 | 0 | 0 | 0 |
| Warm-up Ratio | 0.03 | 0.03 | 0.03 | 0.03 | 0.03 |
| Learning Rate | 3e-5 | 3e-5 | 2e-5 | 3e-5 | 3e-5 |
| Grad Clip | 1.0 | 1.0 | 1.0 | 1.0 | 1.0 |
| SORSA $\gamma$ | 4e-4 | 4e-4 | N/A | 4e-4 | 4e-4 |
| Rank | 128 | 64 | 64 | 64 | 64 |

Table 3: Hyperparameters for evaluation with SORSA, LoRA and PiSSA on different models for GSM-8K and MATH. ML denotes Max Length.

| Model | Llama 2 7B | RWKV6 7B | RWKV6 7B | Mistral 7B | Gemma 7B |
|---|---|---|---|---|---|
| Method | SORSA | SORSA | LoRA & PiSSA | SORSA | SORSA |
| Precision | BF16 | FP32 | FP32 | BF16 | BF16 |
| Sampling | False | False | False | False | False |
| Top-P | 1.0 | 1.0 | 1.0 | 1.0 | 1.0 |
| ML for GSM-8K | 1024 | 1024 | 1024 | 1024 | 1024 |
| ML for MATH | 2048 | 2048 | 2048 | 2048 | 2048 |
| ML for HumanEval | 2048 | 2048 | 2048 | 2048 | 2048 |

## LLM USAGE DISCLOSURE

LLMs were used only to polish language, such as grammar and wording. These models did not contribute to idea creation or writing, and the authors take full responsibility for this paper's content.

Table 4: Hyperparameters of training for with AdaLoRA on different models for GSM-8K and MATH

| Model | Llama 2 7B | Mistral 7B | Gemma 7B | RWKV6 7B |
|---|---|---|---|---|
| Method | AdaLoRA | AdaLoRA | AdaLoRA | AdaLoRA |
| Mix-Precision | TF32+BF16 | TF32+BF16 | TF32+BF16 | TF32+BF16 |
| Epoch | 1 | 1 | 1 | 1 |
| Batch Size | 128 | 128 | 128 | 128 |
| Max Length | 512 | 512 | 512 | 512 |
| Weight Decay | 0 | 0 | 0 | 0 |
| Warm-up Ratio | 0.03 | 0.03 | 0.03 | 0.03 |
| Learning Rate | 2e-5 | 2e-5 | 2e-5 | 2e-5 |
| Grad Clip | 1.0 | 1.0 | 1.0 | 1.0 |
| $\beta_1$ | 0.85 | 0.85 | 0.85 | 0.85 |
| $\beta_2$ | 0.85 | 0.85 | 0.85 | 0.85 |
| $r_{init}$ | 128 | 64 | 64 | 64 |
| $r_{target}$ | 128 | 64 | 64 | 64 |
| $t_{init}$ | 100 | 100 | 100 | 100 |
| $t_{final}$ | 600 | 600 | 600 | 600 |

Table 5: Hyperparameters of evaluation for with AdaLoRA on different models for GSM-8K and MATH. ML denotes Max Length.

| Model | Llama 2 7B | Mistral 7B | Gemma 7B | RWKV6 7B |
|---|---|---|---|---|
| Method | AdaLoRA | AdaLoRA | AdaLoRA | AdaLoRA |
| Precision | BF16 | BF16 | BF16 | FP32 |
| Sampling | False | False | False | False |
| Top-P | 1.0 | 1.0 | 1.0 | 1.0 |
| ML for GSM-8K | 1024 | 1024 | 1024 | 1024 |
| ML for MATH | 2048 | 2048 | 2048 | 2048 |
| ML for HumanEval | 2048 | 2048 | 2048 | 2048 |

