# OpenReview forum: "Orthonormal Regularization in Low-Rank Adaptation"
_ICLR.cc/2026/Conference — Submitted to ICLR 2026_

### Official Review · Reviewer_kNHT · 2025-10-26

**Soundness:** 2
**Presentation:** 1
**Contribution:** 2
**Rating:** 2
**Confidence:** 4

**Summary:**

The paper tackles performance degradation (e.g., catastrophic forgetting) of Parameter-Efficient Fine-Tuning (PEFT) methods (e.g., LoRA) on out-of-domain tasks with limited data, identifying weight matrix condition number as a key factor. It proposes SORSA, a PEFT method that explicitly regularize the condition numbers. Experiments show the effectiveness of the proposed approach.

**Strengths:**

1. The paper is clear written, and the raised problem is interesting.
2. The paper provides some theoretical analysis.

**Weaknesses:**

1. The paper is too rough, e.g., L137.  Some Eqns are not numbered, e.g. L230.
2. The paper should consider a much higher baseline to evaluate the true effectiveness, e.g., [1].
3. The paper claims to suppress the large condition number, but not explicit regs are applied to diag(S).
4. The assumption in Theorem 5.6, "∇Ltrain is invariant" is strange.

[1] Flat-LoRA: Low-Rank Adaptation over a Flat Loss Landscape, ICML'25

**Questions:**

pls see weakness.

---

### Official Review · Reviewer_Sdny · 2025-10-26

**Soundness:** 2
**Presentation:** 2
**Contribution:** 2
**Rating:** 2
**Confidence:** 4

**Summary:**

This paper proposes a PEFT method which tunes only top spectral part with orthonormal conditions. The authors provide theoretic convergence proof and show that the regularization condition helps regularizing the condition number of final tuned weights. Experimental results show the proposed method behaves better than prior baseline methods.

**Strengths:**

The paper connects weight condition number and PEFT, which sounds interesting. Current presentation involves both theoretic side and empirical side.

**Weaknesses:**

1. missing strongly related prior work ([1] and references therein)
2. the experiments are not exhaustive (or just not indicative) enough to showcase the paper's main claim, see question below
3. the motivation is not clear enough, some content in the abstract is not well justified, see question below
4. some typing error, see question below

[1] Spectral Adapter: Fine-Tuning in Spectral Space: https://arxiv.org/abs/2405.13952

**Questions:**

1. in the experiments, only typical benchmarks are presented. However, the authors sell the work as mitigating performance degradation in their abstract. I think adding experiments on fine-tuning on different tasks one after another, and compare the performance degradation is needed.

2. The authors mention "we analyze this issue and identify the condition number of weight matrices as a key factor contributing to such degradation", I don't see justification of this in main content either theoretically or empirically. Why original LoRA will lead to bad condition number?

3. Also, since the authors mention the presented method can help with stablizing weight matrices' condition numbers, I think printing condition numbers of final weights for comparison is needed.

4. how the current method behaves compared to tuning just the top part without orthonormal regularization compared to [1]? An ablation study will be helpful to understand the role of orthonormal regularization.

5. Typing errors:


line 137: it seems the authors want to discuss "Condition Numbers in Neural Networks", but the material is missing except for only a paragraph name.

line 202: the authors mention $\Sigma_p$ to have first $r$ entries  non-zero and $\Sigma_r$ to have first $n-r$ entries non-zero. This seems incorrect. Looks like $\Sigma_p$ should have first $r$ entries  non-zero and $\Sigma_r$ should have first $n-r$ entries zero?

[1] Spectral Adapter: Fine-Tuning in Spectral Space: https://arxiv.org/abs/2405.13952

---

### Official Review · Reviewer_gJaK · 2025-10-27

**Soundness:** 3
**Presentation:** 3
**Contribution:** 2
**Rating:** 4
**Confidence:** 4

**Summary:**

This paper proposes SORSA (Singular Values and Orthonormal Regularized Singular Vectors Adaptation), a novel parameter-efficient fine-tuning (PEFT) approach for large language models. The method builds upon PiSSA, introducing an orthonormal regularization term on the singular vectors to improve the conditioning of adapted weight matrices. The motivation is that LoRA and its variants often amplify the condition number of weights, leading to poor generalization under low-data regimes. The authors provide both theoretical analysis and empirical evaluations across several 7B-scale LLMs (LLaMA-2, RWKV6, Mistral, and Gemma). Experimental results show that SORSA generally outperforms LoRA, PiSSA, and AdaLoRA.

**Strengths:**

* The paper is well-structured and presents a clear theoretical motivation linking matrix conditioning to generalization performance.

* The introduction of an orthonormal regularizer for singular vectors is mathematically sound and theoretically justified with detailed proofs (Theorem 5.4 and 5.6)

**Weaknesses:**

* While the introduction of an orthonormal regularizer on top of PiSSA is conceptually reasonable, the novelty is somewhat limited, and the overall methodological advance may appear incremental.
* The paper lacks analysis on the A and B matrix orthogonality in practice. Without empirical evidence (e.g., orthogonality metrics during training), it is hard to assess whether the proposed regularizer effectively enforces the intended property.
* No analysis of computational overhead (extra FLOPs or training time) is provided; adding regularization and SVD initialization could increase cost.

**Questions:**

* Can the authors show empirical evidence that the orthonormal regularizer indeed increases A, B orthogonality?
* Does the orthonormal regularizer interact with rank selection (r) — e.g., does SORSA benefit more under low ranks compared to higher ones?

**Details Of Ethics Concerns:**

NA.

---

### Official Review · Reviewer_p1iY · 2025-11-05

**Soundness:** 3
**Presentation:** 3
**Contribution:** 3
**Rating:** 4
**Confidence:** 5

**Summary:**

This paper proposes SORSA (Singular Values and Orthonormal Regularized Singular Vectors Adaptation), a novel parameter-efficient fine-tuning method for large language models. The paper identifies the condition number of weight matrices as a key factor contributing to performance degradation on out-of-domain tasks during PEFT. Theoretical analysis includes convergence guarantees and a proof that regularization improves conditioning. Experiments on Llama 2, Mistral, and Gemma demonstrate consistent improvements over LoRA, PiSSA, and AdaLoRA on mathematical reasoning (GSM-8K, MATH) and coding (HumanEval) tasks.

**Strengths:**

- SORSA is simple to implement (similar to PiSSA with added regularizer)
- The link between ill-conditioning and PEFT degradation is intuitive and worth investigating
- Testing on 4 different LLM families (Llama 2, RWKV6, Mistral, Gemma) shows breadth

**Weaknesses:**

- The orthonormal regularizer only encourages orthonormality at convergence, but this does not directly imply improved conditioning. The claim requires showing that gradient steps reduce condition numbers, which is non-trivial and unproven
- SORSA uses learning rate 3e-5 while baselines use 2e-5 (Table 2). The authors justify this as "counterbalancing the negative effect of orthonormal regularizer," but this makes comparisons unfair. it's unclear if improvements come from regularization or simply higher learning rates.
- The paper never measures actual condition numbers during training to verify Theorem 5.6's predictions. Figure 2 shows loss/gradient norms but not κ(W), which is the claimed mechanism.
- Unclear motivation for regularizer choice:  Other regularizers (e.g., nuclear norm, spectral norm constraints) could also improve conditioning. Why is this specific regularization term chosen. Could you provide more justification or comparison?
- Comparision with OLoRA is also missing. It seems like a very closely related work.

Missing details:
- How is γ (regularization strength) chosen? Table 2 shows γ=4e-4, but no ablation or sensitivity analysis is provided.
- The relationship between Equation 2 and Equation 3 is confusing. Why introduce η_d?
- What is "data-whitening SVD" mentioned in Section 6?

**Questions:**

- Why does SORSA help more on RWKV6 on GSM-8K) than on Gemma?
- Why does SORSA underperform on Gemma MATH vs PiSSA?
- No ablation studying the regularizer in isolation (PiSSA + L_reg without separate SVD factors)
- No comparison with other conditioning-improvement techniques (weight normalization, spectral normalization). Could you try a few of these for comparison?
- How exactly does the regularizer L_reg guarantee κ(U^reg_p) < κ(U^unreg_p)?
- What are the actual condition numbers κ(W_p)? Can you plot κ(W_p) over training for SORSA vs PiSSA vs LoRA to validate Theorem 5.6 empirically?
- The motivation discusses catastrophic forgetting, but all experiments test in-domain performance. Can you evaluate pre-trained capabilities (e.g., commonsense reasoning after math fine-tuning)?

---

### Meta-Review · Area_Chair_Jb1q · 2026-01-06

**Summary:**

In this paper, the authors propose to use singular values and orthonormal regularized, termed as SORSA, for parameter-efficient fine-tuning (PEFT). The reviewers’ comments are quite consistent that the novelty is limited, the experiments are not sufficient, and the writing looks rough. The authors did not provide any rebuttal

Overall, the recommendation is REJECTION.

**Reviewer Concerns:**

The reviewers’ comments are generally consistent. (i) the novelty is limited, especially considering a missing reference (pointed out by Reviewer Sdny); (ii) the motivation is not well supported by theoretical discussion or numerical experiments, e.g., the condition number has never been measured; (iii) the baseline is not strong enough; (iv) Some typos have been also pointed out and as Reviewer kNHT said “the paper is too rough”.

**Reviewer Scores:**

The initial score is 2/2/4/4. The authors did not provide any response during the rebuttal period.

---

### Decision · Program_Chairs · 2026-01-26

Reject